# New Insights into Non-Alcoholic Fatty Liver Disease and Coronary Artery Disease: The Liver-Heart Axis

**DOI:** 10.3390/life12081189

**Published:** 2022-08-04

**Authors:** Georgiana-Diana Cazac, Cristina-Mihaela Lăcătușu, Cătălina Mihai, Elena-Daniela Grigorescu, Alina Onofriescu, Bogdan-Mircea Mihai

**Affiliations:** 1Unit of Diabetes, Nutrition and Metabolic Diseases, “Grigore T. Popa” University of Medicine and Pharmacy, 700115 Iași, Romania; 2Clinical Center of Diabetes, Nutrition and Metabolic Diseases, “St. Spiridon” County Clinical Emergency Hospital, 700111 Iași, Romania; 3Institute of Gastroenterology and Hepatology, “Sf. Spiridon” Emergency Hospital, 700111 Iași, Romania; 4Unit of Medical Semiology and Gastroenterology, “Grigore T. Popa” University of Medicine and Pharmacy, 700115 Iasi, Romania

**Keywords:** non-alcoholic fatty liver disease, coronary artery disease, cardiovascular risk, liver-heart axis

## Abstract

Non-alcoholic fatty liver disease (NAFLD) represents the hepatic expression of the metabolic syndrome and is the most prevalent liver disease. NAFLD is associated with liver-related and extrahepatic morbi-mortality. Among extrahepatic complications, cardiovascular disease (CVD) is the primary cause of mortality in patients with NAFLD. The most frequent clinical expression of CVD is the coronary artery disease (CAD). Epidemiological data support a link between CAD and NAFLD, underlain by pathogenic factors, such as the exacerbation of insulin resistance, genetic phenotype, oxidative stress, atherogenic dyslipidemia, pro-inflammatory mediators, and gut microbiota. A thorough assessment of cardiovascular risk and identification of all forms of CVD, especially CAD, are needed in all patients with NAFLD regardless of their metabolic status. Therefore, this narrative review aims to examine the available data on CAD seen in patients with NAFLD, to outline the main directions undertaken by the CVD risk assessment and the multiple putative underlying mechanisms implicated in the relationship between CAD and NAFLD, and to raise awareness about this underestimated association between two major, frequent and severe diseases.

## 1. Introduction

Non-alcoholic fatty liver disease (NAFLD) is a chronic liver disease constantly on the rise among patients with metabolic syndrome (MS), leading to an emerging worldwide epidemic. Therefore, a growing volume of research tries to describe its pathogenesis, to determine the most appropriate therapy, and to identify the most accurate predictors for its evolution and prognosis [1].

Already recognized as the most widespread cause of chronic liver disease around the world, NAFLD is a growing public health problem extending to a prevalence of about 25% in the general population. Well-developed countries display the highest prevalences, due to the current unhealthy, sedentary lifestyle [2,3]. Reports of NAFLD involve more than 50% of persons with type 2 diabetes mellitus (T2DM) and 90% of people with severe obesity [3]. Advanced fibrosis is present in approximately 10–15% of patients with NAFLD in the United States and Europe [3]. Moreover, patients with histologically proven non-alcoholic steatohepatitis (NASH) have an increased risk of liver-related death [3,4]. The global prevalence of NAFLD is expected to increase in line with the rates of obesity and T2DM.

NAFLD is characterized by excessive hepatic accumulation of lipids caused by abdominal obesity and insulin resistance [5]. NAFLD is classically defined by the presence of steatosis affecting more than 5% of all hepatocytes (documented histologically by liver biopsy), in the absence of excessive alcohol consumption and after excluding other causes of steatosis such as drugs, or viral hepatitis, autoimmune diseases, hereditary liver disease or hypothyroidism [5,6]. In a recent move to redefine NAFLD as metabolic dysfunction associated fatty liver disease (MAFLD), criteria in a recent consensus include detection of hepatic steatosis in addition to one of three criteria represented by overweight/obesity, T2DM, or at least two metabolic risk factors [7]. NAFLD encompasses a spectrum of progressive pathological conditions ranging from simple steatosis to NASH, advanced fibrosis, liver cirrhosis, or even hepatocellular carcinoma (HCC) [5,6]. As mentioned above, advanced liver disease due to NAFLD is associated with a worsened prognosis. Therefore, the progression of NAFLD to these late stages of liver histology needs to be prevented.

Since the most important trigger of hepatic insulin resistance is the accumulation of lipids within the liver, NAFLD is perceived today as the hepatic manifestation of the MS [8]. Besides focusing on insulin resistance as the main target of the therapeutic strategy, ongoing studies provide new evidence on additional mechanisms that need to be dealt with in order to intervene on all MS components and to ameliorate all risk factors not only for advanced liver disease, but also for cardiovascular disease (CVD). This constellation of metabolic conditions also includes abdominal obesity, hyperlipidemia, hyperuricemia, chronic kidney disease, and T2DM [9].

Recent epidemiological research has identified a close relationship between these two global health problems, NAFLD and CVD. The progression of NAFLD displays the closest correlation with CVD, followed by extra-hepatic cancers and other liver-associated complications [10]. CVD represents a leading cause of death in the general population, with a prevalence of at least 40% among patients with NAFLD [2,11]. NAFLD is a predictor and a risk factor for the development of CVD, as it increases the risk of morbidity and mortality and impacts the progress to other extrahepatic manifestations [9]. Therefore, the highest mortality in NAFLD seems to be due to the aggravation of the CVD risk driven by metabolic comorbidities, and not to the evolution towards HCC or end-stage liver disease [12,13].

However, a minority of studies have not succeeded to demonstrate this significant association of NAFLD with the risk of CVD. Olubamwo et al. showed that incident CVD can be predicted in patients with NAFLD using a non-invasive test (NIT), but the results were not significant after adjustments based on metabolic factors [14]. Shah and colleagues reported no correlation between hepatic fat and atherosclerotic CVD (ASCVD), demonstrating instead a correlation between the rising incidence of ASCVD and a higher pericardial fat content [15].

In contrast with the few studies that failed to prove that hepatic steatosis significantly contributes to an increased cardiovascular risk, a greater number of researchers demonstrated a link existing between fatty liver and subclinical CVD, independently of traditional cardiovascular risk factors or other MS elements [16,17,18]. Differences in the inclusion criteria and the high variability of tools used for NAFLD and CVD diagnosis may be the explanation of this discrepancy between results [19].

Therefore, there is a high probability that NAFLD is an independent risk factor for CVD, regardless of other associated metabolic risk factors. The multiple conventional risk factors like dyslipidemia, hypertension, obesity, tobacco smoking, and T2DM are also strongly correlated with the incidence and severity of atherosclerosis. A well represented body of evidence supports the hypothesis that atherosclerosis and NAFLD share a handful of common cardiometabolic risk factors [20,21]. NAFLD not only promotes atherosclerosis, but also predisposes to the evolution towards coronary artery disease (CAD), valvular heart disease, left ventricular dysfunction, heart failure, arrhythmia, and stroke [22,23]. A longer duration and the progression to advanced stages of NAFLD are associated with an increased cardiovascular risk, including coronary lesions [24,25], high arterial stiffness [26], impaired endothelial function, or a greater risk for carotid intima-media thickening [27].

Further pathogenesis-related research is needed to identify all mechanisms and then develop targeted therapies able to decrease the additional CVD risk related to NAFLD. The evolution towards cardiovascular events and mortality could be prevented by a sustained screening for cardiovascular risk and an early intervention to reduce it. Since CAD is one of the most frequent forms of ASCVD, such an approach would surely serve to lower, in particular, the prevalence of coronary events and to reduce CAD-related morbidity and death.

In this review, we summarize the current knowledge on the pathogenesis linking together NAFLD and CAD and revise the available evidence validating the hypothesis that these two conditions share key pathological features. We also highlight the results of CAD risk assessment in patients with NAFLD, independently of their metabolic status, and the need for approaches to improve their outcome.

## 2. Epidemiology of the Relationship between NAFLD and CAD

Compared to non-NAFLD individuals, it appears that patients with NAFLD are associated with an important risk of fatal and non-fatal cardiovascular events such as angina, myocardial infarction, coronary revascularization, or stroke [28]. An extensive meta-analysis of six studies carried out on 25,837 adults, including nearly 6000 cases of NAFLD showed that patients with NAFLD had an increased risk of clinical cardiovascular events compared to those without NAFLD (14.9% vs 6.3%) [29].

Narrowing the hypothesis that NAFLD is a risk factor for CVD, the fatty liver also appears to be a risk factor for CAD, independently of common risk factors, such as age, sex, family history of CVD, dyslipidemia, obesity, arterial hypertension, and diabetes [23].

Multiple publications indicate that patients at high risk for both diseases, such as those with diabetes, dyslipidemia, high level of low-density lipoprotein cholesterol, smoking, or family history of CAD, also have a high risk for non-calcified plaques (NCPs) [30,31,32].

A cohort study of 3756 North American individuals evaluated for NAFLD using computed tomography (CT) and for CAD using coronary computed angiography (CCTA) demonstrated that hepatic steatosis is associated with major adverse cardiovascular events (MACE) irrespective of other cardiovascular risk factors or of CAD extent, assessed by measurements of coronary stenosis or plaques [33]. According to Choi et al., the intensity of NAFLD was closely related to the severity of angiography-proven coronary artery stenosis in an Asian population. NAFLD kept its value of CAD predictor independently of common risk factors like age, gender, body mass index (BMI), or glycemic control [34].

In a recent meta-analysis, the prevalence of subclinical CAD in 67,070 patients with NAFLD reached 38.7% (95% confidence interval [CI]: 29.8%–48.5%) of asymptomatic patients (odds ratio [OR]: 1.22; CI: 1.13–1.31, *p* < 0.001), and clinical CAD was present in 55.4% (CI: 39.6%–70.1%) of symptomatic patients (OR: 2.18, CI: 1.69–2.81, *p* < 0.0001); both forms significantly correlated with NAFLD. Non-obstructive CAD had a 43.5% prevalence (CI: 30.3%–57.8%), higher than obstructive CAD, with a 33.5% prevalence (CI: 19.6%–51.1%) [35].

The research conducted by Lee et al. pointed out that only NCPs are independently associated with NAFLD, while the incidence of calcified or mixed plaques did not vary between people with or without NAFLD [36]. NCPs are suggestive of instability and predisposition to acute coronary events, whilst calcified plaques (CPs) add a less vulnerable feature [37]. Considering these findings, the mechanism leading to sudden, unexpected cardiac events in asymptomatic patients with NAFLD may be related to the NCPs instability and the elevated risk of plaque rupture [36].

During NAFLD progression, some data suggests that advanced fibrosis worsens the CAD state. Moreover, NASH has a lower risk for the liver fibrosis stage than for CAD lesions and cardiovascular events [38,39].

Furthermore, research adjusting for cardiometabolic risk factors found NAFLD severity to be independently associated with coronary atherosclerosis, especially with mixed type plaques. Moreover, even the population without associated metabolic risk factors had a higher risk for CAD and mixed atherosclerotic plaques when hepatic steatosis was more severe [40]. A study comparing NASH patients to controls with hepatic steatosis found the former to have a higher risk of coronary lesions (stenosis, NCPs and calcium score) [38].

Another example proving the existence of advanced, high-risk coronary plaques in patients with NAFLD is represented by a cohort study derived from The ROMICAT II trial (Rule Out Myocardial Infarction using Computer Assisted Tomography). Assessment by CCTA and hepatic CT demonstrated an increased prevalence of high-risk plaques compared to patients without NAFLD, irrespective of cardiovascular risk factors and CAD severity. Moreover, NAFLD added an approximately 6-fold higher risk for the development of acute coronary syndromes [41]. The risk of progression from subclinical coronary and carotid atherosclerosis also correlates with NAFLD [42].

Asymptomatic patients with NAFLD submitted to coronary angiography have a higher risk for needing percutaneous coronary interventions or bypass grafting surgery, with an increased risk for fatal and non-fatal outcomes. Among patients with NAFLD meeting the criteria for coronary artery bypass grafting surgery, levels of inflammatory markers were elevated in comparison to patients without NAFLD [43,44].

The prospective and retrospective studies focused on the relationship between NAFLD and clinical and subclinical forms of CAD, are listed in Table 1 and Table 2 (all references are detailed within the tables).

## 3. Screening and Diagnosis

### 3.1. CAD in Patients Assessed for NAFLD

The “gold standard” tool for NAFLD diagnosis and quantification is the liver biopsy, which identifies macrovesicular steatosis. Since it is an invasive method, with many potential complications, biopsy indications are limited to patients with NAFLD at high risk of NASH and/or advanced fibrosis and patients with suspected NAFLD in whom other etiologies of steatosis cannot be ruled out without a liver biopsy [6,89].

The early findings correlating NAFLD and CVD focused on the elevation of liver enzymes. Increased values of serum alanine aminotransferase (ALT) [90], gamma-glutamyl transferase [91], or alkaline phosphatase [92] were associated with other forms of CVD like arterial hypertension and peripheral vascular disease [93].

Non-invasive tests such as clinical scores, serum biomarkers and liver elastographic evaluation provide a good alternative to the diagnosis and staging of NAFLD [89]. According to European guidelines, clinical scores should be used in all patients with NAFLD [5]. A retrospective cross-sectional study involving 34,890 asymptomatic subjects evaluated by ultrasonography (US) analyzed 665 subjects undergoing CCTA imaging. Multiple non-invasive scores, including fatty liver index (FLI), hepatic steatosis index (HSI), Fibrosis-4 score (FIB-4), NAFLD fibrosis score (NFS), Forn’s index, and AST to platelet ratio index (APRI), were applied. Values of NFS, FIB-4, and Forn’s index were higher when coronary artery calcium scoring (CACS) increased. The authors concluded that fibrosis markers incorporating risk factors for CAD demonstrated a good discriminatory power in the prediction of CACS levels over 100 [94].

Another non-invasive test of NAFLD is represented by the liver fat score (LFS), used in a study, including 17,244 participants of the United States National Health and Nutrition Survey (US NHANES) database. LFS showed associations with CAD (adjusted OR: 1.09 per standard deviation [SD], 95% CI: 1.03–1.15, *p* = 0.003), angina (1.08, 1.02–1.13, *p* = 0.005) and congestive heart failure (1.11, 1.04–1.18, *p* = 0.003), but not with myocardial infarction or stroke [95].

Currently available imaging methods for hepatic steatosis and fibrosis are represented by US, liver stiffness measurement (LSM) and controlled attenuation parameter (CAP) by transient elastography (VCTE or FibroScan^®^), CT, magnetic resonance imaging, magnetic resonance spectroscopy (MRS), and magnetic resonance elastography (MRE) [89,96].

Transient elastography is a non-invasive imagistic method that remarked as a clinical tool for staging both liver fibrosis indicated by LSM and hepatic steatosis assessed by CAP, allowing an early detection of NAFLD even in asymptomatic individuals without overt liver disease [97,98]. Use of VCTE measurements in patients from the Framingham Heart Study showed that hepatic fibrosis is significantly correlated with cardiometabolic risk factors represented by high values of BMI and waist circumference, elevated serum transaminases, poor glycemic control, higher systolic and diastolic blood pressure, modified lipid profile [99]. Some studies using FibroScan for the assessment of NAFLD showed an independent link between liver stiffness and CACS [71,100]. A retrospective cohort study highlighted the association between NAFLD and CVD using CAP by transient elastography and found an independent correlation between the degree of steatosis and the incidence of CVD. The results suggest an optimal cut-off CAP value set at 295 dB/m for stratifying the associated CVD risk [101].

The Multi-Ethnic Study of Atherosclerosis (MESA) [102] assessed NAFLD by CT and the cardiovascular risk by CACS in selected patients [15]. Interestingly, these findings identified only pericardial fat, but not fatty liver, to be associated with cardiovascular outcomes, including CAD, calling into question the role of inflammation and insulin resistance.

### 3.2. NAFLD in Patients Assessed for CAD

Current guidelines recommend the assessment of CVD risk in asymptomatic adults using the Framingham Risk Score (FRS), the European SCORE-2 and SCORE-2 OP, and the ASCVD algorithm, each based on the identification of several risk factors [103]. All these models fail to correctly identify NAFLD-related CVD, because features like insulin resistance are not included in the evaluation [95]. The Framingham Risk Score underestimate CVD risk in patients with MS; hence, this category could be subject to a late mitigation of cardiovascular risk. It is therefore reasonable to hypothesize that NAFLD evaluation could help in accurately assessing cardiovascular risk in the general population [95,104].

The CACS is an accurate assessment tool for the presence and development of coronary atherosclerosis, and can be used for monitoring the disease progression, detecting cardiac outcomes, and oversee therapeutic effectiveness [76]. The association between NAFLD and coronary artery calcification has been most frequently demonstrated by calculating the Agatston CACS by CT [26]. Ichikawa et al. propose in their prospective study on patients with T2DM the stratification of NAFLD-associated risk for cardiovascular events using CACS and FRS [64]. Another study including an Austrian screening cohort and using the FRS to assess the CVD risk showed an independent association of NAFLD with CV risk [105].

CCTA is another non-invasive assessment method of coronary arteries that allows their measurement and identification of plaque composition [71]. The lipid-rich coronary plaques are more vulnerable to sudden rupture and predict an increased risk of CV morbi-mortality [53,106]. Increased CACS measured by cardiac multiple detector CT is followed by a negative impact on patients’ outcome, reflecting the total burden of atherosclerosis. NAFLD is related to CACS progression and high-risk plaques, irrespective of traditional CV risk factors [51]. As mentioned in the previous pages, Meyersohn and colleagues showed, using CCTA scan, that the addition of NAFLD to every grade of CAD (no significant CAD, non-obstructive CAD, obstructive CAD) was associated with a higher risk for cardiovascular events than in patients without fatty liver [33].

## 4. Potential Pathogenic Links between NAFLD and CAD

The pathogenic mechanisms associating NAFLD and CAD are still poorly understood. Several mechanisms have been proposed as promoters of both conditions (Figure 1), among which systemic inflammation, gut microbiota, endothelial dysfunction (ED), oxidative stress, or cardiometabolic comorbidities like glucose dysregulation, insulin resistance, dyslipidemia, obesity, and hypertension; all of these usually display a genetic predisposition; a growing line of evidence suggests that the NAFLD–CAD association is tightly linked to dysfunctional secretion of fatty acids, enzymes, cytokine-related anomalies, and pro-atherogenic microRNAs [22,107,108].

Following recent research, MAFLD may be a more appropriate acronym that highlights better the relevant risk factors and NAFLD pathogenesis [109]. The heterogeneity of MAFLD emphasizes the need for individualization, an absolute requisite for the development of new effective treatments for each patient, depending on the dominant subphenotype [110].

Pathogenic features such as insulin resistance, lipid disturbances, oxidative stress, and inflammation can induce NASH and aggravate CVD progression. It is therefore reasonable to suppose that NASH modifications significantly associated with worse coronary artery lesions than hepatic steatosis [40].

### 4.1. Common Risk Factors

The well-established risk factors for CAD are represented by age, gender, family history of premature CVD, hypertension, hyperlipidemia, overweight, T2DM, chronic smoking, or other comorbidities increasing CVD risk [103]. It seems that some residual risk factors remain even after the classical risk factors are mitigated. For example, a study including individuals without classical cardiovascular risk factors showed that even normal levels of low-density lipoprotein (LDL)-cholesterol are associated with subclinical atherosclerosis [111].

The decisive factors leading to NAFLD progression are related to unhealthy lifestyle and eating behavior (excessive intake of saturated fatty acids or fructose, de novo lipogenesis caused by excessive carbohydrate intake), microbiome-related metabolites, and metabolic comorbidities (insulin resistance, dyslipidemia, obesity, T2DM, MS, hypothyroidism) [5,112].

It is therefore obvious that NAFLD and CVD share several common risk factors, e.g., obesity, T2DM, dyslipidemia, and physical inactivity, supporting the idea of a shared pathogenesis [113]. At its turn, insulin resistance is associated both with NAFLD and with endothelial dysfunction and ASCVD [41].

### 4.2. Genetics, Epigenetics Modifications

Also correlating with NAFLD stages, three genetic forms represented by patatin-like phospholipase domain-containing protein-3 (PNPLA3), transmembrane 6 superfamily member 2 (TM6SF2), and sterol regulatory element-binding proteins (SREBP) were found to have a protective effect against CAD [5,95]. The possible negative correlation between PNPLA3 and CAD seems to be influenced by the triglyceride metabolism and NAFLD severity related to PNPLA3 rs738409 mutation [114,115].

CARDIoGRAMplusC4D (Coronary Artery Disease Genomewide Replication and Meta-analysis (CARDIoGRAM) plus the Coronary Artery Disease (C4D)) included a cohort of cases with and without CAD [116]. In this study, TM6SF2 had a protective role for CAD, while the new NAFLD susceptibility gene of the membrane-bound O-acyltransferase domain-containing protein 7 (MBOAT7) had a neutral effect on CAD risk [117].

Other newly identified gene polymorphisms apparently involved in the NAFLD–CAD relationship are represented by: adiponectin rs266729 [118], adiponectin-encoding gene (ADIPOQ), apolipoprotein C3 (APOC3), leptin receptor (LEPR), peroxisome proliferator activated receptors (PPAR), tumor necrosis factor-alpha (TNF-α), microsomal triglyceride transfer protein (MTTP), and manganese superoxide dismutase (MnSOD) [105,119]. The angiotensin (AGT) rs2493132 genotype displayed a significantly increased risk of developing CAD in a Chinese Han population with NAFLD [120]. Rab18 gene expression seems to be linked to increased adiposity and lipotoxicity [121].

Circulating microRNAs are secreted and released in biological fluids and maintain intracellular balance. Metha et al. investigated the expression of microRNAs related to NAFLD and CAD. The researchers found that miR132 circulatory level was reduced in patients with both diseases (0.24 ± 0.16 vs 0.30 ± 0.11, *p* = 0.03) and miR-143 circulatory level was increased compared to controls with NAFLD, but without CAD (0.96 ± 0.90 vs 0.64 ± 0.77, *p* = 0.02). Hence, miRNAs could be utilized as biomarkers to identify and monitor the disease progression [122].

### 4.3. Lipid and Cholesterol Metabolism

Lipid profiles with a pro-atherogenic feature appear to be influenced by the hepatic lipid concentration and the peripheral, adipose insulin resistance; such profiles include an increased proportion of small dense LDL and very low-density lipoproteins (VLDL), high apolipoprotein B to apolipoprotein A-1 ratio, and low high-density lipoprotein (HDL)-cholesterol concentration [6]. Some authors argue that patients with NASH have reduced levels of VLDL due to the decrease of microsomal triglyceride transfer protein and reduced VLDL synthesis. This precursor of the small dense LDL particles that transports an abundance of triglyceride thus becomes a pivotal atherosclerosis risk factor [38]. Prolonged hypertriglyceridemia can determine postprandial hyperlipidemia in patients with NAFLD, which further progresses to an accelerated postprandial atherogenesis and a higher CVD risk [22].

High levels of triglyceride-rich lipoprotein-related elements are related to either calcified or non-calcified coronary lesions in patients with NAFLD [123,124]. Studies such as GREACE (The Greek Atorvastatin and Coronary-heart-disease Evaluation) support the need to prevent major coronary events in patients with elevated plasma liver enzymes caused by NAFLD [125].

Another suggested driver of metabolic and cardiovascular complications seems to be the exhaustion of adipose tissue expansion and ectopic lipid accumulation in non-adipose cells, which in turn causes lipotoxicity [126].

The NAFLD–CVD link is also influenced by lipid profile modifications determined by adipokines like adiponectin, fibroblast growth factor 21 (FGF-21), and adipocyte fatty acid-binding protein (A-FABP) [127,128]. FGF-21 concentrations are elevated in obesity and T2DM, which involves it in NAFLD development. Therefore, the administration of FGF-21 analog can reduce lipogenesis and fatty acid oxidation and can also protect against atherosclerosis progression [129,130]. The association of A-FABP with NAFLD-related CVD is amplified by insulin resistance and arterial inflammation [95]. Fibroblast growth factor 19 (FGF-19) hormone levels were negatively correlated with CAD (defined by coronary angiography), independently of BMI, hypertension, dyslipidemia, and diabetes [131]. Levels of FGF-19 are decreased in patients with obesity, regardless of the degree of insulin resistance. FGF-19 analogs currently under research can suppress *de novo* bile acid synthesis and *de novo* lipogenesis [129,132].

### 4.4. Systemic Inflammation and Cytokines

NAFLD-associated pro-inflammatory status changes the structure of the coronary wall, leading to CAD and increased CVD mortality [133].

The inflammatory syndrome and the increased oxidative stress play a crucial role in CAD associated with NAFLD. Plaque vulnerability is influenced by the inflammatory status of NAFLD. Underlying mechanisms include increased levels of high-sensitive C-reactive protein (hsCRP) and lipoprotein A reported in these patients [40]. Other markers associated both with NAFLD and a high risk for CAD include homeostatic and fibrinolytic function markers, such as fibrinogen, tissue plasminogen activator, and plasminogen activator inhibitor 1 (PAI-1), fetuin-A [55], or homocysteine [50].

The heart-liver axis is related to the MS and acts as a direct connection between the white adipose tissue, the liver and the heart by a systemic signaling led by organic cytokines such as adipokines, hepatokines, and cardiomyokines, predicting the NAFLD-related CVD risk [134]. The adipose tissue produces cytokines with complex outcomes, including a pro-inflammatory effect, such as interleukin 6 (IL-6), interleukin 8 (IL-8), and tumor necrosis factor α (TNF-α). The cumulative pathogenic effects of the disturbed cytokine secretion, the oxidative stress, and the lipotoxicity lead both to NAFLD development and to coronary atherosclerosis, irrespective of conventional cardiovascular risk factors [41,135,136].

Contrary to other studies, Choi et al. did not find insulin, hsCRP, IL-6, and TNF-α levels to be related to CAD; however, the authors found reduced levels serum of adiponectin once the CAD progressed, which indicates a possible dual role that also extends to NAFLD pathogenesis [34]. Adiponectin inhibits hepatic gluconeogenesis and lipogenesis. Therefore, hypoadiponectinemia can determine impaired glucose tolerance, but also CAD in patients without diabetes [137]. In patients with NAFLD, hypoadiponectinemia associates with increased inflammation and oxidative stress. It seems that the early onset of atherosclerosis and CAD is also related to lower serum adiponectin levels, which suggests that hypoadiponectinemia may predict atherosclerosis [138]. The CANTOS (Canakinumab Anti-inflammatory Thrombosis Outcome Study) trial highlighted the key role of interleukin 1 (IL-1), a cytokine related to the evolution of NAFLD, as a therapeutic strategy for atherosclerosis. The results showed a positive outcome on CVD events and morbi-mortality [139].

### 4.5. Endothelial Dysfunction and Oxidative Stress

Endothelial dysfunction developed during NAFLD progression is considered an independent risk factor for CAD occurrence [140]. An impaired coronary flow reserve was described in patients with NAFLD compared to controls without fatty liver after adjustment for cardiometabolic risk factors [141]. An impaired flow-mediated vasodilatation (FMD) can also influence the emergence of vulnerable coronary plaques and the high risk for ischemic heart syndromes in patients with NAFLD [133]. Moreover, NAFLD was associated with a higher short-term mortality and a worsened long-term prognosis in patients with ST-segment elevation myocardial infarction [142].

It was shown that patients with NAFLD may have an impaired endothelial nitric oxide synthase (eNOS) function due to insulin resistance, leading to a reduction in the nitric oxide (NO) substrate production and an imbalance in the induction of platelet-mediated vasorelaxation. Hence, eNOS dysfunction plays a key role in modifying the endothelial function in patients with NAFLD and may determine an increased cardiovascular risk [143].

The role of adipocyte-derived hormone leptin and angiotensin are also investigated for their role in endothelial function as vasoactive factors [144]. Hyperleptinemia in obesity and NAFLD is significantly correlated with the development of atherosclerosis and cardiovascular diseases [145,146]. Stimulation of the renin-angiotensin system and leptin resistance appears to be correlated with arterial hypertension associated with obesity. Moreover, angiotensin II can also participate to the NAFLD pathophysiology by stimulation of lipogenesis, insulin resistance or pro-inflammatory cytokine production [145,147].

Several endothelial biomarkers were studied as possible determinants of the pathophysiological relationship between NAFLD and CAD. Increased endocan levels and decreased levels of high mobility group box 1 (HMGB-1) were correlated with the severity of CAD in NAFLD, while anti-endothelial cell antibodies (AECA) has not yet proven any significance [148,149]. The levels of circulating petraxin-3 (PTX-3), an acute-phase protein, were found to be elevated and strongly correlated with endothelial dysfunction in patients with NAFLD [150].

### 4.6. Gut Microbiota

Compared to healthy individuals, patients with NAFLD, obesity and diabetes display an increased intestinal permeability and increased bacterial growth in the small intestine (endotoxemia) [108]. Metabolic endotoxemia can occur in the form of lipopolysaccharides (LPS) entering portal circulation and impairing the immune response by binding to toll-like receptor 4 (TLR) and activating the inflammatory cascade [108,151]. This process acts on the insulin signaling, favors hepatic steatosis and progression to NASH; on the other hand, it promotes endothelial dysfunction, LDL oxidation, and thrombogenesis, destabilizing the atherosclerotic plaques [152].

Gut dysbiosis was discovered in patients with both CAD and NAFLD. The intestinal microbiota might be different in patients with NAFLD and CAD than in those with just CAD. Studies focused on gut microbiota composition in patients with NAFLD and CAD [152] showed increased levels of *Coprococcus* and *Veillonella,* and decreased levels of Bacteroides fragilis, *Parabacterioides*, *Bifidobacterium longum* subsp. *infantis*, *Ruminococcus gnavus, Bacteroides dorei*, which could underlie the intestinal alterations that cause a higher risk for adverse CVD outcomes less witnessed in NAFLD-free CAD. The abundance of *Coprococcus* could favor MS in patients with CAD and NAFLD due to its positive correlation with BMI [153]. Another potential cause of the progression of NAFLD and CAD could be the abundance of *Collinsella* and *Proteobacteria* [154].

Circulating bile acids (BA) are also implicated in metabolic liver diseases associated with CVD. Glycochenodeoxycholic acid (GCDCA), a marker for reduced serum concentrations of BA, predicts CAD. Interestingly, this defect is reversible under statin therapy [155].

Gut microbiome-related metabolites, such as phosphatidylcholine (PC) and trimethylamine N-oxide (TMAO), are also studied for their association with an increased cardiovascular risk [152]. TMAO is a metabolite linked with the PC metabolism and modulates glucose and lipid homeostasis, thus influencing the liver, precipitating intra-adipose inflammation and impairing platelet function. High levels of TMAO seem to be involved in the progression of NAFLD-related CAD, probably due to intestinal dysbiosis influenced by dietary factors [105]. TMAO was found able to predict cardiovascular events in a cohort of patients submitted to coronary angiography, independently of other risk factors [156]. Among current attempts for therapeutic strategies aiming at gut microbiota in CAD, an inhibitor targeting a pair of microbial TMA-generating enzymes was developed, which was able to reduce the risk of atherothrombotic events and prevent coronary complications [157].

## 5. The Challenge of Lean NAFLD and Cardiovascular Risk

Patients with NAFLD usually are overweight or obese and associate insulin resistance, T2DM, dyslipidemia, hypertriglyceridemia, or hypertension, all of these being MS components and CVD risk factors [105]. However, it appears that CVD risk is increased even in individuals with NAFLD but normal BMI, who became categorized as lean patients with NAFLD (BMI <25 kg/m^2^ in Caucasians and <23 kg/m^2^ in Asians) [158]. The prevalence of lean NAFLD ranges from 10% to 20%, and despite the absence of obesity these individuals have a similar cardiovascular risk to patients with obese NAFLD [159].

In lean NAFLD, the risk of cardiometabolic conditions is elevated compared to NAFLD-free subjects of all BMI categories [160]. When lean NAFLD was compared with a healthy control group, an increased prevalence of metabolic impairment and cardiovascular risk was noticed [161]. An analysis on 5375 lean participants selected from the NHANES III survey showed that NAFLD presence was associated with a major increase in all-cause and cardiovascular mortality compared to controls [162].

While most opinions support an important role for obesity in patients with both NAFLD and CVD, some recent studies suggest a new theory of a particular lean NAFLD phenotype displaying a higher CVD risk than overweight people [163]. This hypothesis suggests that visceral adiposity has a higher contribution to the waist circumference value in lean NAFLD persons; this ectopic adipose accumulation leads to endothelial dysfunction and a pro-inflammatory effect [164]. Visceral fat accumulation in lean Asian people was correlated with the severity of NAFLD [95].

Despite the seemingly favorable metabolic risk profile, lean NAFLD is associated with a higher rate of cardiovascular events compared to the obese NAFLD group, as shown in a recent subgroup analysis study [165]. Likewise, another retrospective *post hoc* analysis showed the risk for incident CVD of various types (CAD, ischemic stroke, and cerebral hemorrhage) in lean patients with NAFLD to be higher than in overweight patients with NAFLD (8.8% vs. 3.3%) [166]. Therefore, NAFLD also needs an increased attention in lean individuals to prevent cardiovascular events.

The differences between lean and obese NAFLD include genetic predisposition, body composition, environmental risk factors, and gut microbiota, all related to the incidence of cardiovascular disease [167,168]. The metabolic dysfunction in NAFLD is weight-dependent. A recent meta-analysis showed lean people with NAFLD have significantly lower values of systolic and diastolic blood pressure and fasting glycemia than patients with NAFLD and obesity [169]. According to a study by Kim and colleagues, lean patients with NAFLD had a significantly higher ASCVD score (defined as an ASCVD risk of >10%) compared to patients with NAFLD and obesity (51.6% vs. 39.8%) and NAFLD-free controls (25.5%) [170]. NAFLD influences the incidence of CVD more than the presence of any degree excess weight, indicating that NAFLD-triggered mechanisms favor ASCVD independently of overweight or obesity [166].

The impact of lean NAFLD on the long-term prognosis of such patients is not completely understood, but it could be labeled as not being a benign condition [171]. The fact that NAFLD is also found in normal weight patients is usually overlooked, delaying the diagnosis and risking the progression of hepatic steatosis to NASH or fibrosis, with an increased associated CVD risk [172]. The conundrum of lean NAFLD being linked to CVD risk needs clarification in future studies.

At this moment, no data are available on the incidence and progression of coronary artery disease in lean NAFLD.

## 6. Conclusions

The available findings strongly support the fact that NAFLD and CAD are two conditions closely related to the MS. Similar to NAFLD being named the hepatic MS manifestation, we could say that CAD is its cardiac manifestation, closely related to the former. Consistent data showed that CAD has a high prevalence among patients with NAFLD, leading to an increased mortality. NAFLD is significantly associated with clinical and subclinical CAD, independently of the conventional cardiometabolic risk factors.

Many putative mechanisms are considered relevant in NALFD-related CAD, including genetics, inflammation, oxidative stress, lipotoxicity, atherogenic dyslipidemia, or gut microbiota. Key questions for future research refer to the complex mechanisms linking NAFLD to CAD, to the nature of optimal personalized lifestyle modification and appropriate pharmacologic approaches for both conditions, and to whether NAFLD-directed therapeutic strategies can also reduce CVD risk.

## Figures and Tables

**Figure 1 life-12-01189-f001:**
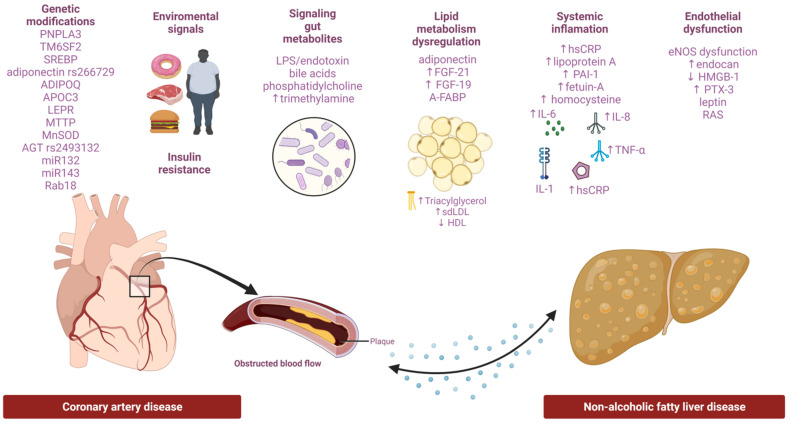
Summary of suggested pathophysiological mechanisms underlying the NAFLD–CAD interconnection. *Abbreviations:* PNPLA3, patatin-like phospholipase domain-containing protein-3; TM6SF2, transmembrane 6 superfamily member 2; SREBP, sterol regulatory element-binding proteins; ADIPOQ, adiponectin-encoding gene, APOC3, apolipoprotein C3; LEPR, leptin receptor; MTTP, microsomal triglyceride transfer protein; MnSOD, manganese superoxide dismutase; AGT, angiotensin; LPS, lipopolysaccharides; FGF-21, fibroblast growth factor-21; FGF-19, fibroblast growth factor-19; A-FABP, adipocyte fatty acid-binding protein; hsCRP, high-sensitive C-protein reaction; PAI-1, plasminogen activator inhibitor-1; IL-1, interleukin-1; IL-6, interleukin-6; IL-8, interleukin-8; TNF- α, tumor necrosis factor-alpha; HMGB-1, high mobility group box 1; PTX-3. petraxin-3; RAS, renin-angiotensin system.

**Table 1 life-12-01189-t001:** Summary of studies that evaluated the association between NAFLD and clinical CAD.

Author, Year, Ref.	Country	Study Type	NAFLD Diagnosis	CAD Diagnosis	Patients Characteristics	Impact of NAFLD on CAD/ Results
**Thévenot et al., 2022** **[45]**	France	Prospective (CORONASH)	NIT FibroScan	Coronary angiography	189	5.3% advanced liver fibrosis (LSM ≥ 8 kPa) eLIFT, NFS—good sensitivity and specificity as first-line screening test for liver fibrosis
**Hsu et al., 2021** **[40]**	Taiwan	Retrospective	US APRI	CCTA	1502 893 NAFLD 581 CAD	Steatosis severity associated with mixed plaque pattern (*p* = 0.043)
**Fiorentino et al., 2020** **[46]**	Italy	Retrospective	US	Coronary angiography	1254 601 NAFLD 130 CAD	prediabetes and NAFLD—increased risk of CVD or CAD by 2.3 and 2 fold T2DM with NAFLD—2.3 and 2 fold higher risk of CVD or CAD
**Niikura et al., 2020** **[38]**	Japan	Prospective	Liver biopsy	CCTA CACS (CT)	101 NAFLD 51 CACS	NASH and fibrosis—independent RF for CAS NASH—not significantly associated with presence of CACS NASH independent RF for high-risk plaque
**Seba et al., 2020** **[47]**	India	Prospective	US FibroScan	Coronary angiography SINTAX Score	300 CAD 165 NAFLD	NAFLD associated with CAD No correlation between NAFLD grades and CAD
**Liu HH et al., 2019** **[48]**	China	Prospective	US	Coronary angiography	162 CAD 40 NAFLD	NAFLD—independent predictor of CVD outcomes in patients with stable, new-onset CAD (OR: 2.72, 95% CI: 1.16–6.39, *p* = 0.022)
**Langroudi TF et al., 2018** **[49]**	Iran	Retrospective	US	Coronary angiography	264 191 NAFLD 127 CHD	NAFLD presence and grade not correlated with coronary arteries ATS and its severity in non-diabetic patients
**Pulimaddi et al., 2016** **[50]**	India	Cross-sectional	US	ECG/coronary angiogram/angioplasty	150 T2DM >30 years	59.3% prevalence of CAD in the NAFLD group (significant statistically)
**Sinn et al., 2017** **[51]**	South Korea	Retrospective	US	CACS	4731 2088 NAFLD	NAFLD significantly associated with the development of CAC independent of CV and metabolic RF
**Idilman et al., 2015** **[52]**	Turkey	Retrospective	CT	CCTA	273 T2DM 59 NAFLD 44 CAD	NAFLD—associated with CAD in T2DM *p* = 0.04
**Osawa K et al., 2015** **[53]**	Japan	Retrospective	CT	CT	414 64 NAFLD 22 CHD	NAFLD—independent predictor of high-risk plaques (OR: 4.60; 95% CI: 1.94–9.07, *p* < 0.01
**Puchner SB et al., 2014** **[41]**	USA	Prospective	CT	CCTA	445 205 CP 190 NCP	NAFLD—significantly associated with the presence of high-risk plaque (adjusted OR: 2.13; 95%, CI: 1.18, 3.85), adjusted for CV RF and the extent and severity of CAD
**Agaç et al., 2013** **[54]**	Turkey	Prospective	US	Coronary angiography	80, acute coronary syndrome	81.2% patients with NAFLD and acute coronary syndrome; NAFLD associated with higher SYNTAX score (OR: 13.20; 95% CI: 2.52–69.15)
**Ballestri S et al., 2014** **[55]**	Italy	Retrospective	US Fetuin-A	Coronary angiography	29 NAFLD 20 CAD	High Fetuin-A associated with NAFLD and lower risk of CAD
**Choi DH et al., 2013** **[34]**	South Korea	Prospective	US	Coronary angiography	134	NAFLD—independent predictor for CAD (*p* = 0.03, OR: 1.685; 95% CI: 1.051–2.702); Increased proportion of severe fatty liver in higher grade CAD; Adiponectin level decreased once the CAD progressed
**Josef et al., 2013** **[56]**	Israel	Retrospective	CT	CCTA	29 NAFLD 9 CHD	Smaller retinal AVR (<0.7)—increased risk for CAD and carotid atherosclerosis in NAFLD even without hypertension or diabetes
**Wong VW-S et al., 2011** **[57]**	Hong Kong	Prospective	US	Coronary angiography	612 356 NAFLD 301 CAD	Steatosis (adjusted OR: 2.31; 95% CI: 1.46–3.64) and alanine aminotransferase level (adjusted OR: 1.01; 95% CI: 1.00–1.02) independently associated with CAD
**Assy et al., 2010** **[58]**	Israel	Prospective	CT	CT	29 NAFLD 11 CHD	NAFLD—associated with high prevalence of CP and NCP, independently of the MS and CRP
**A****ç****ikel M et al., 2009** **[59]**	Turkey	Retrospective	US	Coronary angiography	355 215 NAFLD 153 CHD	NAFLD—independent predictor of CHD (> 50% stenosis of ≥1 major coronary artery) after adjustment for CVD risk factors
**Arslan U et al., 2007** **[60]**	Turkey	Retrospective	US	Coronary angiography	65 NAFLD 39 CHD	NAFLD—independent predictor of CHD (>50% stenosis of ≥1 major coronary artery) after adjustment for CVD risk factors and MS

**Table 2 life-12-01189-t002:** Summary of studies that evaluated the association between NAFLD and subclinical CAD.

Author, Year, Ref.	Country	Study Type	NAFLD Diagnosis	CAD Diagnosis	Patients Characteristics	Impact of NAFLD on CAD/Results
**Carter et al., 2022** **[61]**	Scotland	Post-hoc analysis of Prospective Scottish Computed Tomography of HEART trial	CT	CT (CACS)	1726 155 hepatic steatosis	Hepatic steatosis associated with increased prevalence of CAD No difference in MI in those with and without steatosis (1.9% vs. 2.4%, *p* = 0.92)
**Ichikawa et al., 2022** **[62]**	Japan	Prospective	CT	CCTA	1148 247 hepatic steatosis 977 suspected CAD	High association between hepatic steatosis and increased risk of MACE in suspected stable CAD
**Wang X et al., 2022** **[63]**	China	Retrospective	FIB-4 score	Coronary angiography Gensini score	342 105 NAFLD	NAFLD severity—associated with CAS High FIB-4 score—high CAC
**Chen et al., 2021** **[25]**	Taiwan	Prospective	US	CACS (CT)	545 437 NAFLD 242 CAC	1.36-fold greater risk of developing CAC in patients with different severity of NAFLD vs. those without NAFLD (OR: 1.36, 95% CI: 1.07–1.77, *p* = 0.016)
**Ichikawa et al., 2021** **[64]**	Japan	Prospective	CT	CACS FRS	529 T2DM	NAFLD, CACS, and FRS-associated with CVE (HR and 95% CI: 5.43, 2.82–10.44, *p* < 0.001; 1.56, 1.32–1.86, *p* < 0.001; 1.23, 1.08–1.39, *p* = 0.001, respectively)
**Meyersohn NM et al., 2021** **[33]**	North America	Nested cohort study	CT	CCTA	3756	Hepatic steatosis associated with MACE (4.4% vs 2.6% in those without steatosis) indepently of other CV RF/extent of CAD
**Saraya et al., 2021** **[65]**	Egypt	Prospective	CT	CCTA	800 440 CAD	NAFLD and high-risk plaque features: Napkin ring sign, Positive remodeling, Low HU, and Spotty calcium (OR: 7.88, 95% CI (4.39–14.12), *p* < 0.001, OR: 5.84, 95% (3.85–8.85), *p* < 0.001, OR: 7.25, 95% CI (3.31–15.90), *p* < 0.001 and OR: 6.66, 95% CI (3.75–11.82), *p* < 0.001)
**Bae YS et al., 2020** **[66]**	South Korea	Retrospective	US NFS, FIB-4 index	CCTA	3693 244 CAS 1588 NAFLD	NAFLD associated with CAS (≥50% stenosis) stronger in women, but absolute risk higher in men
**Ismael H et al., 2020** **[19]**	Egypt	Prospective	FibroScan	Coronary angiography Gensini score	100 42 NAFLD	S2-S3 NAFLD and CVD (OR: 24, 95% CI: 17–31)
**Koo BK et al., 2020** **[67]**	USA	Retrospective	CT	CCTA	719 NAFLD 443 CHD	NAFLD significantly associated with coronary calcification (OR: 1.28; 95% CI: 1.07–1.53)
**Chang Y et al., 2019** **[68]**	South Korea	Retrospective	US FIB-4 score, APRI	CACS	105328 34382 NAFLD 5249 CAD	NAFLD, AFLD associated with CAC
**Oni E et al., 2019** **[69]**	USA	Retrospective	CT	CACS CIMT	4123 729 NAFLD 386 CHD	NAFLD—independently associated with CAC> 0 and CIMT > 1 mm
**Pais et al., 2019** **[70]**	France	Retrospective	FLI	FRS CACS (CT)	2617 930 NAFLD	High prevalence of CAC (183 ± 425 vs 117 ± 288, *p* < 0.001) in those with hepatic steatosis vs without
**Park HE et al., 2019** **[71]**	South Korea	Retrospective	CAP	CCTA Coronary plaque >1.5 mm^2^	330 NAFLD 186 CAD 147 NCP	CAP-defined NAFLD significantly associated with NCP, independent with cardiometabolic RF (adjusted OR: 3.528, 95% CI: 1.463–8.511, *p* = 0.005), no significant correlation with CP (*p* = 0.171)
**Sinn DH et al., 2019** **[51]**	South Korea	Retrospective	US NFS	Hospitalization for MI	111492 37263 NAFLD 183 MI	NAFLD associated with increased incidence of MI independent of RF
**Gummesson et al., 2018** **[72]**	Sweden	Retrospective	CT	CACS (CT)	106 NAFLD 73 CHD	NAFLD and CACS association in subjects with few other metabolic risk factors (60% subjects of the total cohort) with 0 or 1 of the 7 predefined RF; OR: 5.94, 95% CI: 2.13 ± 16.6
**Lee SB et al., 2018** **[36]**	South Korea	Retrospective	US, FLI, NFS	CCTA	5121 38.6% NAFLD	NAFLD associated with NCP; significant association of FLI ≥30 with NCP (1.37, 95% CI: 1.14–1.65, *p* = 0.001) and NFS ≥ −1.455 with NCP (1.20, 95% CI: 1.08–1.42, *p* = 0.030)
**Wu R et al., 2017** **[73]**	China	Retrospective	US	CACS (CT)	2345 1272 NAFLD 237 CHD	NAFLD—significantly associated with the development of coronary artery calcifications (adjusted OR: 1.348, 95% CI: 1.030–1.765)
**Jacobs K et al., 2016** **[74]**	USA	Retrospective	US	CACS (CT) (VAT)	250 71 NAFLD 52 CHD	NAFLD and CAC—no clear association Increased CAC, VAT with age, but no increased NAFLD
**Kim JB et al., 2016** **[75]**	South Korea	Retrospective	US	CT EFV	1472 677 NAFLD 147 CHD	Higher EFV levels and NAFLD prevalence in individuals with MS than those without MS (81.0 cm^3^ vs 57.3 cm^3^, *p* < 0.001; 75.6% vs 36.5%, *p* < 0.001)
**Park HE et al., 2016** **[76]**	South Korea	Retrospective	US	CCTA (CAC)	1732 846 NAFLD 413 CAC	NAFLD associated with CAC development independent of other metabolic RF in those without CAC at baseline, but not with CAC progression in those with CAC at baseline DM risk factor for CAC progression
**Al Rifai M et al., 2015** **[18]**	USA	Retrospective	CT	CACS (CT)	3976 670 NAFLD 362 CAC	NAFLD—associated with inflammation and CAC
**Kim MK et al., 2015** **[77]**	South Korea	Retrospective	US	CACS (CT)	919 Postmenopausal women 294 NAFLD 81 CAC	OR for prevalence of CAC: no NAFLD, 1.0; mild NAFLD, 1.34 (95% CI: 0.92–2.16); moderate to severe NAFLD, 1.83 (95% CI: 1.06–3.16) NAFLD—not independent factor for CAD in postmenopausal women
**Kang MK et al., 2015** **[78]**	South Korea	Retrospective	US	CT	346 NAFLD 173 CHD	NAFLD—associated with coronary plaques OR: 1.48; 95% CI: 1.05–2.08, *p* = 0.025
**Lee M-K et al., 2015** **[79]**	South Korea	Retrospective	US	CACS (CT)	10063 NAFLD 1843 CAD 340 CACS>100	NAFLD relatively increased risk for CAC vs non-NAFLD; higher OR than that in subjects with abdominal obesity [1.360; 95% CI: 1.253–1.476) vs (1.220; 95% CI: 1.122–1.326)]
**Efe D et al., 2014** **[80]**	Turkey	Retrospective	CT	CT	372 204 NAFLD 107 CAD	Higher prevalence of CAD in NAFLD than non-NAFLD
**VanWagner et al., 2014** **[81]**	USA	Retrospective	CT	CACS (CT)	2424 232 NAFLD 88 CAD	Increased CAC (37.9% vs 26.0%, *p* < 0.001) in NAFLD cases Obesity attenuates NAFLD-ATS relation
**Chhabra et al., 2013** **[82]**	USA	Retrospective	CT	CACS (CT)	400 43 NAFLD 15 CAD	Hepatic steatosis—independent predictor of CACS
**Juarez-Rojas et al., 2013** **[83]**	Mexico	Retrospective	CT	CACS (CT)	765 163 NAFLD 64 CHD	Fatty liver associated with T2DM and MS
**Khashper et al., 2013** **[84]**	Israel	Retrospective	CT	CACS (CT)	318 93 NAFLD 70 CAD	Increased VAT in patients with coronary artery plaques, *p* < 0.001
**Sung KC et al., 2013** **[85]**	South Korea	Retrospective	US	CACS (CT)	7371 39.5% NAFLD 4.5% CACS > 0	Steatosis and baPWV are independently associated with the presence of CAC
**Arslan et al., 2012** **[86]**	Turkey	Prospective	US	Coronary angiography	151 98 NAFLD	64.9% patients with NAFLD NAFLD associated with poor coronary collateral development
**Kim D et al., 2012** **[26]**	South Korea	Prospective	US	CACS (CT)	4023 1617 NAFLD 649 CAD	High CACS significantly associated with the presence of NAFLD (OR: 1.28, 95% CI: 1.04–1.59, *p* = 0.023) independent of visceral adiposity
**Sung KC et al., 2012** **[87]**	South Korea	Retrospective	US	CACS (CT)	3784 NAFLD 510 CAD	Steatosis (OR: 1.21, 95% CI: 1.01–1.45, *p* = 0.04) and HOMA-IR (1.10; 1.02–1.18, *p* = 0.02) associated with CACS > 0
**Agarwal et al., 2011** **[88]**	India	Prospective	US	CIMT	124 T2DM 71 NAFLD 43 CAD	60.5% CAD of the patients with NAFLD; 45.2% of the ones without NAFLD NAFLD—risk marker for CAD in T2DM

*Abbreviations:* NAFLD, non-alcoholic fatty liver disease; NASH, non-alcoholic steatohepatitis; CAD, coronary artery disease; CHD, coronary heart disease; CCTA, coronary computed tomography angiography; CAC, coronary artery calcification; CACS, coronary artery calcium score; CT, computed tomography; FLI, Fatty Liver Index; NFS, NAFLD Fibrosis Score; US, ultrasonography; ECG, electrocardiogram; T2DM, type 2 diabetes; MS, metabolic syndrome; CP, calcified plaques; NCP, non-calcified plaques; ATS, atherosclerosis; CAS, coronary artery stenosis; CV, cardiovascular; CVE, cardiovascular events; CVD, cardiovascular disease; MI, myocardial infarction; RF, risk factor; LSM, liver stiffness measurement; APRI, AST to platelet ratio index; baPWV, brachial-ankle pulse wave velocity; CIMT, carotid intima-media tissue; VAT, visceral abdominal adipose tissue; EFV, epicardial fat volume; FRS, Framingham score; VAT, visceral adipose tissue; CRP, C-reactive protein; HOMA-IR, homeostatic model assessment for insulin resistance; HR, hazard ratio; OR, odd ratio; CI, confidence interval.

## Data Availability

Not applicable.

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
