# Peer review of "New Insights into Non-Alcoholic Fatty Liver Disease and Coronary Artery Disease: The Liver-Heart Axis"

_life, 2022, doi:10.3390/life12081189_

Round 1

Reviewer 1 Report

GENERAL COMMENTS

The topic is interesting as it provides important insight into the links between NAFLD and CVD. Given the high prevalence of these pathologies its clinical relevance is more than justified. The tables are very welcome to summarize a huge amount of information. However, some aspects might be improved.

The manuscript may benefit from considering the following aspects:

Use people first language throughout the manuscript – replace “obese individuals/patients/subjects, etc” by “individuals/patients with obesity”. Also the term “morbidly” referring to obesity is outdated and should be replaced by “severe obesity”.

Page 1, at the beginning of the Introduction: suggest to include a recent reference to support the statement like e.g. Francque SM, Marchesini G, Kautz A, et al. Non-alcoholic fatty liver disease: A patient guideline. JHEP Rep. 2021 Sep 17;3(5):100322. 

The recent call to redefine NAFLD to metabolic associated fatty liver disease (MAFLD) focusing on obesity and metabolic dysfunction should be mentioned. 

The role of transient elastography (Fibroscan ®), by assessing the controlled attenuation parameter (CAP), as a non-invasive method able to accurately assess the presence and the grade of steatosis in stratifying cardiovascular (CV) risk should be mentioned in more detail (ref de Sousa Magalhães R, Xavier S, Magalhães J, Rosa B, Marinho C, Cotter J. Transient elastography through controlled attenuated parameter assisting the stratification of cardiovascular disease risk in NAFLD patients. Clin Res Hepatol Gastroenterol. 2021 Sep;45(5):101580. doi: 10.1016/j.clinre.2020.11.010. Epub 2020 Dec 2. PMID: 33279452).

The allostatic hypothesis might be referred to in the context of NAFLD. The adipose tissue expandability hypothesis states that a failure in the capacity for adipose tissue expansion, rather than obesity per se is the key factor linking positive energy balance and type 2 diabetes (ref Virtue S, Vidal-Puig A. Adipose tissue expandability, lipotoxicity and the Metabolic Syndrome--an allostatic perspective. Biochim Biophys Acta. 2010 Mar;1801(3):338-49). Ectopic lipid accumulation in non-adipocyte cells causes lipotoxic insults including insulin resistance, apoptosis and inflammation.

The relevance of body composition, in general, and of visceral adiposity, in particular, should be outlined since it exerts a decisive impact on the hepato-cardio axis, which in fact, is an adipo-hepato-cardio axis. The adipose tissue is extarordinarily active and produces and secretes multiple adipokines affecting homeostasis. The links between adipokines, inflammation, and lipotoxicity should be mentioned (refs Frühbeck G, Gómez-Ambrosi J. Control of body weight: a physiologic and transgenic perspective. Diabetologia. 2003 Feb;46(2):143-72  //  Pulido MR, Diaz-Ruiz A, Jiménez-Gómez Y, et al. Rab18 dynamics in adipocytes in relation to lipogenesis, lipolysis and obesity. PLoS One. 2011;6(7):e22931). Moreover, some of these adipokines do actually exert important roles as vasoactive factors like leptin and angiotensin (ref Fortuño A, Rodríguez A, Gómez-Ambrosi J, et al. Leptin inhibits angiotensin II-induced intracellular calcium increase and vasoconstriction in the rat aorta. Endocrinology. 2002 Sep;143(9):3555-60) or as hepatokines like the fibroblast growth factor sor FGFs (refs Gómez-Ambrosi J, Gallego-Escuredo JM, Catalán V, et al. FGF19 and FGF21 serum concentrations in human obesity and type 2 diabetes behave differently after diet- or surgically-induced weight loss. Clin Nutr. 2017 Jun;36(3):861-868  //  Alvarez-Sola G, Uriarte I, Latasa MU, et al. Fibroblast growth factor 15/19 (FGF15/19) protects from diet-induced hepatic steatosis: development of an FGF19-based chimeric molecule to promote fatty liver regeneration. Gut. 2017 Oct;66(10):1818-1828). This is especially important also for the case in lean individuals according to BMI that, however, are overweight or even obese when looking at their body fat.

In point 5 about the challenges of NAFLD and CVR in lean the very recent article of Tang A, Ng CH, Phang PH, Chan KE, Chin YH, Fu CE, Zeng RW, Xiao J, Hao Tan DJ, Quek J, Lim WH, Mak LY, Wang JW, Chew NW, Syn N, Huang DQ, Siddiqui MS, Sanyal A, Muthiah M, Noureddin M. Comparative Burden of Metabolic Dysfunction in Lean NAFLD vs. Non-Lean NAFLD - A Systematic Review and Meta-Analysis. Clin Gastroenterol Hepatol. 2022 Jul 18:S1542-3565(22)00669-3. doi: 10.1016/j.cgh.2022.06.029. Epub ahead of print. PMID: 35863685. Should be mentioned.

 A figure trying to schematically summarize the main factors would be certainly very useful to illustrate the content.

Reviewer 2 Report

This well-written manuscript is a comprehensive review, which focuses on the association between non-alcoholic fatty liver disease and coronary artery disease. The topic of the review is clinically important and the relevant findings of the literature are discussed in a logical order. I have only a few minor comments.

Specific suggestions:

     1.  Due to the current preference in scientific literature of a non-stigmatizing language to describe diseases, adjectives should be avoided. Therefore, e.g. instead of "NAFLD patients", "patients with NAFLD" should be written. Please, amend this in lanes 43, 44, 80, 150, 159, 170, 172, 340, 393, 408, 420, 449, 465, 469, and 476.

     2.  IL-6 cannot be considered just as a pro-inflammatory cytokine, it exerts several beneficial effect in the entire body.

     3. L49: “NAFLD” instead of “non-alcoholic fatty liver disease”

     4. L105-106, 175-177, 324-325: The statements require references.

     5. L160: “NCPs” instead of “non-calcified plaques”

     6. L228: “ASCVD” instead of “Atherosclerotic Cardiovascular Disease (ASCVD)”

     7. L232: “MS” instead of “metabolic syndrome”

     8. L398: “nitric oxide (NO) substrate production” instead of “nitric oxide substrate production (NO)”

     9. L436: “seem to be involved” instead of “seem involved”
